# Potential effects on cardiometabolic risk factors and body composition by short message service (SMS)-guided training after recent minor stroke or transient ischaemic attack: post hoc analyses of the STROKEWALK randomised controlled trial

Birgit Maria Vahlberg [ID],[1] Erik Lundström,[2] Staffan Eriksson,[3,4] Ulf Holmback,[1] Tommy Cederholm[1]

For numbered affiliations see end of article.

**Correspondence to**
Dr Birgit Maria Vahlberg;
birgit.vahlberg@pubcare.uu.se

## ABSTRACT

**Objectives** To evaluate effects of mobile phone text-messaging exercise instructions on body composition, cardiometabolic risk markers and self-reported health at 3 months after stroke.

**Design** Randomised controlled intervention study with per-protocol analyses.

**Setting** University Hospital in Sweden.

**Participants** Seventy-nine patients (mean (SD) age 64 (10) years, 37% female) ≥18 years with good motor function (modified Rankin Scale ≤2) and capable to perform 6 min walking test at hospital discharge were randomised to either intervention (n=40) or control group (n=39). Key exclusion criteria: subarachnoid bleeding, uncontrolled hypertension, severe psychiatric problems or cognitive limitations.

**Interventions** The intervention group received beyond standard care, daily mobile phone instructional text messages to perform regular outdoor walking and functional leg exercises. The control group received standard care.

**Main outcome measures** Fat mass and fat-free mass were estimated by bioelectric impedance analysis. Cardiometabolic risk factors like blood lipids, glycated haemoglobin and blood glucose were analysed at baseline and after 3 months.

**Results** Both groups changed favourably in fat-free mass (1.83 kg, 95% CI 0.77 to 2.89; p=0.01, effect size (ES)=0.63 vs 1.22 kg, 95% CI 0.39 to 2.0; p=0.05, ES=0.54) and fat mass (−1.30 kg, 95% CI −2.45 to −0.14; p=0.029, ES=0.41 vs −0.76 kg, 95% CI −1.74 to 0.22; p=0.123, ES=0.28). Also, many cholesterol related biomarkers improved; for example, total cholesterol −0.65 mmol/L, 95% CI −1.10 to −0.2; p=0.06, ES: 0.5 vs −1.1 mmol/L, 95% CI −1.47 to −0.56; p>0.001, ES=0.8. However, there were no between-group differences. At 3 months, 94% and 86%, respectively, reported very good/fairly good health in the text messaging and control groups.

### Strengths and limitations of this study

► This is a randomised controlled clinical trial with an intervention that could be performed anywhere and without extended cost for staff.

► The paper presents secondary analysis using data from a previous randomised controlled trial.

► The post hoc study was not powered for changes in body composition and cardiovascular risk-markers in blood.

**Conclusions** No clear effect of 3 months daily mobile phone delivered training instructions was detected on body composition, cardiovascular biochemical risk factors or self-perceived health. Further research is needed to evaluate secondary prevention efforts in larger populations after recent stroke.

**Trial registration number** NCT02902367.

## INTRODUCTION

Patients with recent stroke and transient ischaemic attacks (TIAs) are at high risk for another vascular event that might lead to permanent disability or death.[1] Three out four strokes can be attributed to behavioural risk factors such as smoking, unhealthy diets and sedentary lifestyles. TIA and stroke share the same risk factors.[1 2] A systematic review revealed that people after stroke are highly sedentary, in particular during the acute phase,[3] which may affect body composition negatively.[1 4 5] Loss of muscle mass and increase of fat mass may contribute to insulin resistance and impaired glucose control.[4] Beyond increasing muscle mass and

reduction of fat mass,[5 6] physical activity and exercise may improve lipid and glucose metabolism.

Physical activity and exercising is of paramount clinical concern, and current recommendations[7] advise people to engage in regular physical exercise of moderate intensity for ≥30 min/day.[1 8] The optimal approach to facilitate physical activity for secondary prevention after stroke or TIA is not clear.[6 9 10] The use of mobile phones in healthcare is gaining popularity and many apps are available.[11] A simple strategy such as short message service (SMS) for optimising home-based exercising shows potential to be implemented in clinical practice.[11] We have recently showed improvements in walking endurance and lower body strength after a 3-month mobile-phone guided training programme.[12]

The importance of body composition and the role of muscle mass in metabolic regulation is recently highlighted.[8 13] Muscle mass is lost and fat mass increased during ageing that may result in sarcopenia or sarcopenic obesity.[13] A systematic review summarises the effects of exercise-based interventions on traditional cardiovascular risk factors, showing improvements of blood pressure, fasting glucose and fasting insulin, and increasing high-density lipoprotein (HDL) cholesterol in patients after stroke or TIA.[6] Whether interventions involving self-training after receiving training instructions via SMS could improve cardiometabolic biochemical risk factors, or the distribution of body fat and muscles mass has not been studied. Beyond showing improvements in walking endurance and lower body strength, we wanted to analyse this area in more detail.[12] In this post hoc analysis, we explored whether effects from the text messaging supported training could induce improvements in body composition and cardiometabolic risk. Thus, the aims of this study in patients after recent stroke or TIA were to investigate whether daily mobile phone text messaging coupled with the use of a training diary and a pedometer were better than current standard treatment to improve body composition and cardiometabolic risk markers and self-reported health at 3 months after stroke.

## METHODS
The STROKEWALK study was a single-centre, parallel-group, randomised controlled trial performed at the stroke unit at the Uppsala University Hospital, Sweden. We used the Consolidated Standards of Reporting Trials (CONSORT) checklist when writing our report.[14]

Design, recruitment, determination of sample size, inclusion criteria and intervention have been described in detail elsewhere.[12] In summary, inclusion criteria were age 18 years or older with minor stroke (infarction or intracerebral haemorrhage; first-ever or recurrent) or TIA; with cognition and motor capacity, for example, walking endurance, enough to perform the 6 min walking test (with or without walking aid).

### Procedure
Recruitment started on November 2016, and the last follow-up assessment was performed on December 2018. Patients were evaluated by one physiotherapist (BMV), and a research assistant performed the body composition measurements at median 5 days after verified event. All participants gave oral and written informed consent to be included in the study.

Study participants underwent a physical examination, laboratory tests and answered questionnaires. Demographic and comorbidity data were collected from the medical records. Blood pressure was registered as the last measurement before leaving hospital. We calculated the Charlson Comorbidity Index—an instrument that predicts 10 year survival[15]—to classify comorbidity. Information on self-reported health after 3 months and 1 year were obtained from the Swedish quality Riksstroke registry. Education level was divided into university studies or not. Smokers were grouped into daily-smokers and non-smokers. Ex-smokers were categorised as non-smokers.

### Randomisation
The allocation to either intervention or control group was based on randomisation, 1:1 ratio, stratified by gender. The random allocation sequence is described more thoroughly in the primary outcome paper.[12]

### Baseline examinations
Examinations at baseline have been extensively described previously.[10] In summary we used:

► The Montreal Cognitive Assessment scale (0–30 points) to evaluate cognitive function.[16]
► The modified Rankin Scale (range 0 (no symptoms) to 6 (death)).[17]
► The 6 min walking test (metres) as a submaximal exercise test.[18] Participants walked for 6 min over a 30 m course.
► Stress profile: The questionnaire refers to stress behaviours in everyday life and the stress profile was assessed by a four-level self-reported scale. The scale combines 20 claims with agreements expressed in terms of time urgency/impatience or easily aroused irritation/hostility; for example, other people's mistake irritate me.[19] It is scored from not at all to fully agree, 0–3.
► The four-level Saltin Grimby Physical Activity Level Scale was used to measure self-reported leisure-time physical activity during the past year before stroke.[20]
► The European Working group on Sarcopenia in Older People-2 definition for sarcopenia; that is, probable sarcopenia is indicated by low muscle strength by the five-time chair-stand test >15 s or handgrip strength (Jamar dynamometer)<27 kg in men and <16 kg in women, respectively.[13] Sarcopenia was confirmed by fat-free mass index (see below)<17 kg/m$^2$ for men and 15 kg/m$^2$ for women.

## Outcome assessments for post hoc analyses

The study included several endpoints at 3 months, that is, body composition changes in fat-free mass and fat mass by bioelectric impedance analyses, and biochemical cardiometabolic risk factors and nutritional status.

The fat-free mass (kg) and fat mass (kg) were measured with the bioelectric impedance analyses device Inbody S20; Inbody, Seoul, South Korea. Measurements were performed in the supine position. Fat-free mass index and fat mass index was calculated by dividing masses in kg by height (m) squared.

Body mass index (BMI) was calculated as body weight (kg) divided by height (m) squared. The participants were divided into categories: underweight (<18.5 kg/m²), normal weight (18.5–24.9 kg/m²), overweight (25–29.9 kg/m²), obese class I (30–35 kg/m²), class II (>35–39.9 kg/m²) and class III (≥40 kg/m²).[1]

Non-fasting venous blood samples for baseline analyses of glycated haemoglobin (HbA1c), serum insulin-like growth factor and biochemical cardiometabolic risk factors were obtained. The samples were frozen at −70°C until analyses. The biochemical analyses were performed by accredited methods of the Clinical Chemistry Laboratory at the Uppsala University Hospital, Uppsala.

The cardiometabolic biochemical risk factors total, LDL and HDL cholesterol, and triglycerides were analysed. Baseline values of blood lipids and C reactive protein were recorded from values obtained at admission to the emergency ward or the day after admission to the stroke unit. A C reactive protein level of >5 mg/L was considered to indicate ongoing inflammation.

Non-fasting blood samples at 3 months follow-up were collected and frozen at −70°C until analyses. The serum lipid concentrations were considered pathological when the non-HDL cholesterol (total cholesterol minus HDL cholesterol) was >2.2 mmol/L; LDL cholesterol >1.4 mmol/L and HDL cholesterol was ≤1.2 mmol/L in women and ≤1.1 mmol/L in men,[8] respectively.

Self-reported health at 3 months and 12 months was analysed from data retrieved from the Riksstroke national registry. Reported answers were very good, fairly good, fairly bad, very bad health. Data were only available for the patients with stroke, not for patients with TIA.

Mortality at 1 year was checked by reviewing the individual's medical records that is connected to the official death records.

## Text-messaging intervention

As previously described, the intervention was an add-on to standard care and comprised three different strategies: (1) daily text-messaging (SMS) with instructions what and how to exercise, (2) training diaries for 3 months and (3) pedometers for step counts registration at the first and last week of the study.[12]

The daily morning text messages across the study period gave instructions on how to exercise to increase walking endurance and lower body strength. There was no possibility to text back for help or advice. The training programme was initiated about 1 week after the patients had been randomised.

## Control intervention

Patients in the control group were given standard post-stroke care, which usually does not include specific advice about physical activity or home exercising. The control participants had no restrictions regarding physical activity, exercise or taking part in rehabilitation services.

## Statistical analyses

No a priori power analysis was performed in the present post hoc study and statistical calculations were based on per protocol analysis. Data and variables were checked for missing values and potential outliers.

Categorical data are presented as percentage and descriptive data as mean and SD or median and IQR. To check for normal distribution, the Shapiro-Wilk W test and histogram viewing were used.

Significance of between group differences in physical activity at baseline were analysed by Pearson's ($\chi^2$ test with Fisher's exact test for nominal data or the Kruskal-Wallis test for non-normal or ordinal data. One-way Analysis of variance (ANOVA) tests with Bonferroni corrections were used to analyse continuous normal-distributed data.

Differences in changes between baseline and the 3-month follow-up between the text-messaging group and the control group are presented with 95% CI and tested by non-parametric methods (Mann-Whitney U test) for ordinal or non-normally distributed variables. For all other data, parametric methods (the independent student's t-test) were used.

Within group differences; that is, the difference between baseline and the 3 months follow-up were tested using the paired samples t-test or the Wilcoxon's signed rank test. Correlation strength was calculated by Spearman's r coefficient.

The SPSS, V.27, was used for the analyses (SPSS). The level of statistical significance was set at p<0.05.

## RESULTS
### Baseline characteristics

Figure 1 summarises the recruitment and participant flow in a CONSORT diagram. Out of 100 individuals that were eligible for the study, 79 were invited to participate. A total of 71 individuals completed the full study period; that is, a retention rate of 90%. Table 1 gives baseline characteristics of the text messaging and control groups. Fifteen per cent of the included individuals had type 2 diabetes mellitus. At baseline, a total of 11 individuals (15%) had a C reactive protein level >5 mg/L. At baseline, individuals with HbA1C≥42 mmol/mol were significantly more sedentary compared with those with moderate to high physical activity, as measured with the Saltin Grimby Physical Activity Level Scale.

Table 2 displays baseline characteristics with subjects divided into groups according to the Saltin-Grimby

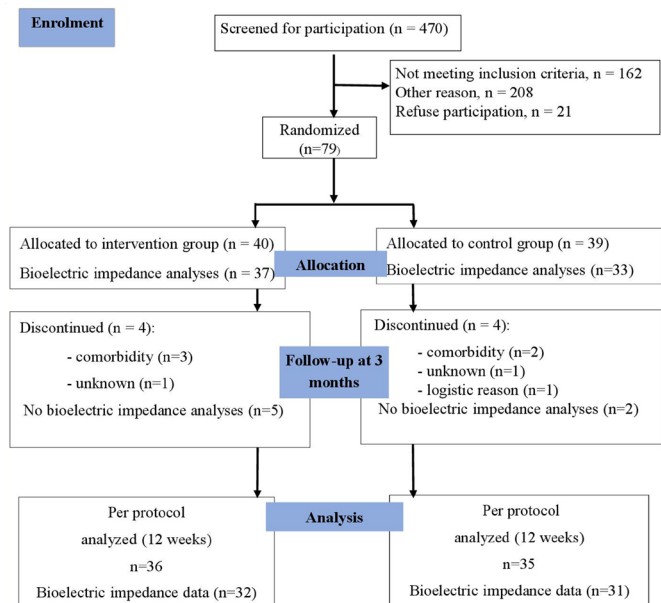

**Figure 1** CONSORT diagraming. Flow chart of the study participants through the phases of the original randomised controlled trial. Data are collected in Uppsala University Hospital, Sweden (November 2016–December 2018). CONSORT, Consolidated Standards of Reporting Trials.

Physical Activity Level Scale at baseline. Sedentary participants compared with those with moderate/high physical activity had an overall worse risk-factor profile with significant differences; for example, higher body mass index, HbA1c, fat mass and a lower HDL cholesterol. Missing data are reported.

The body mass index range was 20.1–42.8 kg/m². At baseline, 15% had probable sarcopenia according to either a reduced chair stand test or reduced handgrip strength. None had confirmed sarcopenia, that is, reduced lower body strength combined with low muscle mass.

### Changes of body composition and cardiometabolic risk markers during the intervention period

Table 3 shows that on average there was no weight change in neither the text-messaging group nor in the control group at 3 months, whereas variability in weight changes in both groups were substantial; that is, from −10.1 to 7.8 kg. A positive within group effect on HDL cholesterol was seen in female participants that received daily text messages, but there were no between-group effects.

Fat-free mass increased significantly in both groups, with no difference between groups (table 3). Concurrently, fat mass decreased in both groups, but again no difference between groups was observed.

Table 3 shows that total cholesterol, LDL and HDL cholesterol changed in beneficial directions over the 3-month period irrespective of group allocation, but no significant difference between groups were found.

Only minor changes were observed for HbA1c in both groups. Serum insulin-like growth factor tended to decrease in both groups, but no significant difference

**Table 1** Baseline characteristics of individuals allocated to the text messaging or control group

| | Text messaging (N=40) | Control group (N=39) |
|---|---|---|
| Sex, male | 26 (67) | 24 (60) |
| Age (years), mean (SD) | 63.9 (10) | 63.9 (10) |
| Type of stroke, | | |
| Cerebral infarction | 29 (72) | 28 (72) |
| Intracerebral haemorrhage | 4 (10) | 5 (13) |
| Transient ischaemic attack | 7 (18) | 6 (15) |
| Lifestyle factors | | |
| Saltin Grimby Physical Activity Level Scale, score | | |
| Sedentary | 2 (5) | 9 (23) |
| Light physical activity | 29 (73) | 24 (62) |
| Moderate/high physical activity | 9 (22) | 6 (15) |
| Body mass index (kg/m²), mean SD | 27.3 (4.2) | 27.7 (4.9) |
| Normal weight <25 | 12 (30.0) | 12 (30.8) |
| Overweight, 25–29.9 | 17 (42.5) | 17 (43.6) |
| Obese, class I 30–35 | 9 (22.5) | 6 (15.4) |
| Obese, class II>35–39.9 | 2 (5.0) | 3 (7.7) |
| Obese, class III≥40 | 0 | 1 (2.6) |
| Stress profile, score, | | |
| <25 (not stressed) | 27 (68) | 25(64) |
| 25–34 (sometimes stressed) | 11(27) | 7 (18) |
| ≥35 (highly stressed) | 2 (5) | 7 (18) |
| Diabetes mellitus type 2 | 5 (12) | 7 (18) |
| Glycated haemoglobin, mmol/mol, mean (SD) | 37.6 (7.6) | 38.4 (7.6) |
| Non-diabetic | 36.4 (4.2) | 35.3 (3.4) |
| Systolic blood pressure, mean (SD) | 129.1 (14.5) | 132.2 (18.5) |
| >140 | 4 (10) | 11 (30) |
| Diastolic blood pressure, mean (SD) | 75.1 (11.4) | 77.0 (10.7) |
| >90 | 2 (5) | 3 (8) |

Data are given as n (%) if not otherwise stated.

between groups were found. Although only 37% of the participants were women, we performed sex-specific subgroup analyses of body composition variables as well as of biochemical cardiometabolic risk markers, which did not change the results.

At baseline, C reactive protein was not associated with any of the cardiovascular risk-markers nor fat-free mass, fat mass or self-reported physical activity level.

### Mortality, self-reported health and adverse events

One participant of the control group had passed away at 1 year. Data on self-reported health at 3 months and 1 year after the event could be retrieved in 62 and 50 of all participating individuals with stroke. No difference

**Table 2** Level of physical activity at baseline, reported from the Saltin Grimby Physical Activity Scale

| | Missing values n (%) | All | Sedentary | Light physical activity | Moderate/high physical activity | P value |
|---|---|---|---|---|---|---|
| n (%) | | 79 (100) | 11 (14) | 53 (68) | 15 (19) | |
| Women, n (%) | 0 | 29 (37) | 2 (18) | 24 (45) | 3 (20) | 0.75 |
| Age, yrs, mean (SD) | 0 | 63.9 (10.4) | 61.8 (9.0) | 64.0 (11.3) | 64.9 (8.0) | 0.45 |
| Diabetes mellitus, yes n (%) | 0 | 12 (15) | 3 (27) | 8 (15) | 1 (7) | 0.79 |
| University studies, yes n (%) | 0 | 40 (62) | 5 (46) | 26 (49) | 9 (60) | 0.087 |
| Non-smoker, n (%) | 0 | 71 (90) | 8 (73) | 48 (91) | 15 (100) | 0.08 |
| High intake fruit and vegetables, n (%) | 2 (2.5 %) | 55 (70) | 6 (54) | 36 (69) | 13 (93) | |
| Body weight, kg, median (IQR) | 0 | 80.90 (25.1) | 97.80 (28.60) | 80.90 (22.90) | 77.3 (11.40) | **0.027** |
| Body mass index, kg/m², median (IQR) | 0 | 26.90 (5.70) | 30.90 (6.50) | 27.0 (5.15) | 24.90 (2.4) | **0.016** |
| Fat-free mass, kg, mean (SD) | 9 (11.4%) | 57.61 (11.10) | 64.18 (10.86) | 55.72 (11.31) | 58.92 (9.45) | 0.08 |
| Fat-free mass index, kg/m², mean (SD) | 9 (11.4%) | 18.91 (2.14) | 20.0 (2.24) | 18.75 (2.16) | 18.64 (1.92) | 0.20 |
| Fat-free mass per cent, mean (SD) | 9 (11.4%) | 70 (10) | 68 (10) | 69 (9.8) | 74 (9.9) | 0.21 |
| Fat-mass, kg, mean (SD) | 9 (11.4%) | 25.10 (10.9) | 32.0 (13.0) | 24.92 (10.35) | 18.96 (8.47) | **0.042** |
| Fat-mass index, kg/m², mean (SD) | 9 (11.4%) | 8.40 (3.69) | 10.0 (4.13) | 8.58 (3.63) | 6.79 (3.13) | 0.087 |
| Fat- mass per cent, mean (SD) | 9 (11.4%) | 30 (10) | 32 (10) | 30 (9.8) | 26 (9.8) | 0.20 |
| Systolic blood pressure, (mm Hg), mean (SD) | 4 (5%) | 131 (16.6) | 137 (18.9) | 129 (15.6) | 132 (17.7) | 0.29 |
| Diastolic blood pressure, (mm Hg), median (IQR) | 4 (5%) | 75 (15) | 80 (10) | 70 (10) | 85 (20) | 0.13 |
| P-cholesterol, (mmol/l), mean (SD) | 3 (3.8%) | 5.13 (1.20) | 4.52 (1.17) | 5.20 (1.19) | 5.33 (1.19) | 0.18 |
| P-LDL cholesterol, (mmol/l), median (IQR) | 3 (3.8%) | 3.20 (1.78) | 2.60 (1.50) | 3.20 (1.80) | 3.60 (2.13) | 0.61 |
| P-HDL cholesterol, (mmol/l), median (IQR) | 3 (3.8%) | 1.30 (0.50) | 0.99 (0.37) | 1.30 (0.60) | 1.35 (0.42) | **0.018** |
| Women, mean (SD) | | 1.48 (0.42) | 1.11 (0.27) | 1.51 (0.44) | 1.55 (0.07) | |
| Men, median (IQR) | | 1.20 (0.43) | 0.99 (0.33) | 1.20 (0.43) | 1.25 (0.45) | |
| P-LDL/HDL ratio, (mmol/L), mean (SD) | 3 (3.8%) | 2.57 (1.05) | 2.86 (1.11) | 2.46 (1.04) | 2.71 (1.07) | 0.45 |
| P-triglycerides, (mmol/l), median (IQR) | 3 (3.8%) | 1.22 (0.62) | 1.49 (0.45) | 1.23 (0.64) | 1.11 (0.91) | 0.70 |
| P-glucose, (mmol/l), median (IQR) | 13 (16.4%) | 6.90 (1.70) | 7.60 (2.0) | 6.65 (1.60) | 6.90 (1.70) | 0.55 |
| P-HbA1c, (mmol/mol), median (IQR) | 3 (3.8%) | 36.0 (5.0) | 39.0 (26.0) | 35.0 (6.0) | 35.0 (5.0) | 0.053 |
| P-C reactive protein, (mg/L), median (IQR) | 6 (7.6%) | 1.70 (2.3) | 1.40 (1.63) | 2.00 (2.44) | 1.10 (3.15) | 0.51 |
| S-insulin-like growth factor 1 (µL), mean (SD) | 13 (16.4%) | 161.8 (41.0) | 189.9 (57.0) | 156.7 (36.4) | 160.8 (40.2) | 0.11 |

Significance of between group differences was analysed by Pearson's χ2 test with Fisher's exact test for nominal data or the Kruskal-Wallis test for non-normal or ordinal data and the one-way ANOVA test with Bonferroni correction for continuous normal distributed data. The significance level was set at p<0.05.

The significance level (in bold) was set at p<0.05.

ANOVA, Analysis of variance; HbA1c, glycated haemoglobin; HDL, high-density lipoprotein cholesterol; LDL, low density lipoprotein cholesterol.

**Table 3** Baseline and follow-up outcome of cardiometabolic risk factors and body composition by group, analysed as per protocol

| | Text messaging group | | | | Control group | | | | Mean/median difference (95% CI) | P value |
|---|---|---|---|---|---|---|---|---|---|---|
| | n | Baseline | 3 months | P value | n | Baseline | 3 months | P value | | |
| Women | 16 | | | | 13 | | | | | |
| Body weight, kg, median (IQR) | 33 | 81.9 (15.2) | 82.4 (15.2) | 0.28 | 31 | 83.3 (30.1) | 83.7 (17.7) | 0.43 | −0.2 (−1.40 to 0.8) | 0.45 |
| Body mass index, kg/m² median (IQR) | 33 | 26.6 (6.0) | 26.36 (6.0) | 0.69 | 31 | 27.30 (5.9) | 26.36 (6.0) | 0.98 | −0.026 (−0.47 to 0.36) | 0.36 |
| Fat-free mass, kg mean (SD) | 32 | 57.2 (9.4) | 59.0 (10.3) | **0.001** | 31 | 58.65 (12.9) | 59.86 (13.1) | **0.005** | −0.62 (−1.9 to 0.70) | 0.35 |
| Fat-free mass%, mean (SD) | 32 | 71 (10) | 72 (10) | **0.014** | 31 | 71 (10) | 72 (10) | **0.029** | −0.006 (−0.21 to 0.009) | 0.43 |
| Fat-free mass index, kg/m², mean (SD) | 32 | 18.93 (1.9) | 19.52 (2.0) | **0.001** | 31 | 19.00 (2.4) | 19.41 (2.5) | **0.004** | −0.18 (−0.6 to 0.24) | 0.39 |
| Fat mass, kg, mean (SD) | 32 | 24.69 (11.5) | 23.39 (11.2) | **0.029** | 31 | 24.65 (10.9) | 23.89 (10.0) | 0.123 | 0.74 (−0.95 to 2.0) | 0.48 |
| Fat mass%, mean (SD) | 32 | 29.3 (10) | 27.6 (10) | **0.012** | 31 | 29.1 (10) | 28.2 (9.6) | **0.041** | 0.007 (−0.008 to 0.022) | 0.33 |
| Fat mass index, kg/m², mean (SD) | 32 | 8.30 (3.9) | 7.89 (3.8) | **0.036** | 31 | 8.19 (3.7) | 7.93 (3.4) | 0.103 | −0.16 (−0.64 to 0.33) | 0.52 |
| P-cholesterol (mmol/l) | 35 | 5.13 (1.19) | 4.47 (1.0) | **0.006** | 32 | 5.11 (1.23) | 4.09 (1.15) | **<0.001** | −0.36 (−0.99 to 0.26) | 0.25 |
| P-non-HDL cholesterol (mmol/L), mean (SD) | 34 | 3.83 (1.16) | 2.95 (1.00) | **<0.001** | 32 | 3.83 (1.0) | 2.65 (0.95) | **<0.001** | −0.29 (−0.89 to 0.31) | 0.34 |
| P-LDL cholesterol, (mmol/l), mean (SD) | 34 | 3.17 (1.15) | 2.37 (0.88) | **<0.001** | 33 | 3.13 (1.0) | 2.37 (0.80) | **<0.001** | −0.30 (−0.83 to 0.24) | 0.28 |
| P-HDL cholesterol. (mmol/l), mean (SD) | 34 | 1.32 (0.33) | 1.53 (0.41) | **<0.001** | 33 | 1.30 (0.48) | 1.45 (0.50) | **<0.001** | −0.03 (−0.13 to 0.07) | 0.55 |
| Women | 13 | 1.42 (0.37) | 1.69 (0.44) | **0.001** | 10 | 1.56 (0.57) | 1.61 (0.58) | 0.24 | −0.16 (−0.35 to 0.03) | 0.10 |
| Men | 21 | 1.26 (0.30) | 1.43 (0.37) | **0.003** | 23 | 1.18 (0.40) | 1.38 (0.50) | **<0.001** | 0.037 (−0.08 to 0.15) | 0.54 |
| P-LDL/HDL cholesterol, ratio, mean (SD) | 34 | 2.54 (1.00) | 1.66 (0.72) | **<0.001** | 32 | 2.64 (1.05) | 1.53 (0.74) | **<0.001** | −0.23 (−0.67 to 0.21) | 0.29 |
| Glycated haemoglobin, (mmol/mol), median (IQR) | 35 | 35.5 (5.0) | 37.0 (5.8) | 0.067 | 32 | 36.0 (5.0) | 37.0 (5.0) | 0.11 | 0 (−1 to 1) | 0.81 |
| S-Insulin-like growth factor-1 (µ/L), mean (SD) | 29 | 167.9 (45.2) | 152.8 (52.8) | 0.061 | 33 | 157.6 (37.6) | 149.6 (41.4) | 0.086 | 7.0 (−10.36 to 24.44) | 0.42 |

All within group values are presented as mean (SD). Significance of between group differences in glycated haemoglobin, weight and body mass index was analysed by the Mann-Whitney U test. The median estimate was calculated using the Hodges-Lehmann test. Within-group differences for those variables are analysed with Wilcoxon signed-ranks test.

Significance of between-group differences in all variables except for glycated haemoglobin, weight and body mass index was analysed by the independent samples t-test. Within group differences was analysed by the paired samples t-test.

The significance level (in bold) was set at p<0.05.

Data are presented as means (SD) or median (IQR).

HDL, high-density lipoprotein cholesterol; LDL, low-density lipoprotein cholesterol.

between the groups was registered; that is, at 3 months 94% and 86%, respectively, reported very good or fairly good health in the text messaging and control groups. At 1 year, 89% and 86%, respectively, reported very good or fairly good health in the text-messaging and control groups.

No adverse events such as fall-related fractures, syncope or dizziness events requiring hospitalisation until 3 months follow-up were reported.

## Patient and public involvement

Patients were involved in the decision on how to deliver the training instructions; that is, by text messages or video links by SMS. In a pilot study,[21] we investigated whether patients after stroke preferred text SMS or SMS with video (You-Tube) instructions, with a preference for the first approach. Otherwise, there was no direct patient or public involvement.

## DISCUSSION

This study is a post hoc analysis of the STROKEWALK study, where a text-messaging intervention was previously reported to increase the walking capacity as well as chair-stand capacity, that is, the primary outcomes of the study.[12] However, the post hoc analyses showed that beneficial effects on body composition and some biochemical cardiometabolic risk factors were achieved in both groups, irrespective if they received text messages or not. Thus, the positive effects on walking capacity in the intervention group did not transform to beneficial effects on body composition or the selected group of biochemical cardiometabolic risk factors in the intervention group only.

It is well known that weight reduction and regular physical exercise improve insulin sensitivity and decreases triglyceride levels.[8] Aerobic physical activity such as 25–30 km brisk walking per week may increase HDL cholesterol by 0.08–0.15 mmol/L.[8] Overweight and especially abdominal adiposity contributes to overall dyslipidaemia and these conditions were common in this study population that also showed both excessive weight loss and weight gain after 3 months. Unfortunately, we were not able to control for fluctuation in weight or depressive symptoms that became present during the study period. Both the text-messaging group and the control group increased their fat-free mass, whereas fat mass decreased only in the text messaging intervention group. A similar positive within group effect on HDL cholesterol was observed in female participants that received daily text messages, which is in line with the positive effects on walking and lower body strength after the training intervention.

A Cochrane review concluded that there is currently limited high quality research to support lifestyle interventions targeting mortality, blood pressure, lipid profiles and insulin resistance after stroke or TIA.[22] Treatment targets for extended secondary prevention[8] for body weight could be body mass index $<25 \text{ kg/m}^2$, moderately vigorous physical activity for 30–60 min most days, and for individuals with diabetes HbA1c <53 mmol/mol.

In recent years, the importance of muscle mass for mobility, muscle strength, energy, lipid and glucose metabolism has been highlighted in rehabilitation.[13 23] Probable sarcopenia, that is, low muscle strength and confirmed sarcopenia, that is, the addition of low muscle mass is gaining increasing interest as risk factors for falls and fractures. Impaired activities of daily living are associated with cardiac and respiratory disease, as well as to cognitive impairment.[13] At baseline, 12 participants displayed probable sarcopenia. Thus, interventions to improve muscle function are important in a population that have suffered from stroke or TIA. Many already have a low level of physical activity before the cerebrovascular event, and after the event the activity level might become even lower if no actions are taken.

For some individuals the walking programme and the number of repetitions of chair-rising might have been too low to increase muscle mass. For this purpose, individualised exercise instructions with higher intensity might be needed. There are not always opportunities for supervised exercise for those with minor stroke or TIA when discharged from the hospital. Therefore, methods including tele-rehabilitation and self-training can be an alternative or complement to usual rehabilitation, especially in remote or underserved areas.[24]

As this was an explorative post hoc study, power was not calculated for the outcomes in this study. It is obvious that this post hoc study might have been underpowered to show any differences in body composition or cardiometabolic risk factors. The study population of the current study showed great variability in many measures, which may have implications for the necessary sample size. The effects of exercise are known to be strongest in individuals that are sedentary.[25] Approximately 14% of the participants in this study were sedentary at study start. By chance 9 of the 11 sedentary individuals were randomised to the control group, which might have had an impact on the results. Other potential reasons for the weak effects could be that the time period was too short to induce stronger effects, or the substantial fluctuations in weight.

In conclusion, the results of this study should be interpreted with caution, since they are the outcome of secondary analyses of a previously reported randomised controlled trial. We observed that body composition; that is, fat-free mass and fat mass, as well as some cardiometabolic risk factors like various cholesterol fractions, improved slightly 3 months after a stroke or TIA both in the text-messaging group and in the control group. A majority of the participants reported good or fairly good health after 3 months. The study indicates that not a negligible amount of individuals after stroke have cardiovascular risk-factors that need to be considered for in secondary prevention.

## Clinical messages

► Body composition, HDL and LDL cholesterol improved in poststroke patients after 3 months irrespective of the intervention.

► Additional studies are needed in larger study populations over longer study periods to elucidate if potential cardiometabolic beneficial effects could be achieved.

► Especially sedentary individuals need help how to increase physical activity and reduce risk of recurrent cerebrovascular events.

**Author affiliations**
[1]Department of Public Health and Caring Sciences, Clinical Nutrition and Metabolism, Uppsala Universitet Medicinska fakulteten, Uppsala, Sweden
[2]Department of Neuroscience, Uppsala Universitet Medicinska fakulteten, Uppsala, Sweden
[3]Centre for Clinical Research, Eskilstuna, Sweden
[4]Department of Community Medicine and Rehabilitation, Physiotherapy, Umeå University, Umeå, Sweden

**Collaborators** We thank research assistant Mia Berglund for the co-operation in the execution of the study. We also thank Marina Lahti (www.intime.se) for assisting with sending the text messages.

**Contributors** This study was conceived, organised and managed by BMV, EL, UH, SE and TC. All authors listed above contributed to the study design and data interpretation. Writing of the first draft of the paper was done by BMV and all authors were involved in preparation and critique of the manuscript. The Uppsala Research Center assisted with some statistical analyses. BV acts as a guarantor of the study.

**Funding** This study was funded by the Medical Faculty at Uppsala University, the Swedish Stroke Association (Stroke-Riksförbundet), the Geriatric funding and the Swedish Associations of Physiotherapists, Neurology.

**Competing interests** None declared.

**Patient and public involvement** Patients and/or the public were involved in the design, or conduct, or reporting, or dissemination plans of this research. Refer to the Methods section for further details.

**Patient consent for publication** Not applicable.

**Ethics approval** Ethical approval was obtained from the regional Ethical Review Board of Uppsala University Hospital, Sweden: Dnr: 2015/550.

**Provenance and peer review** Not commissioned; externally peer reviewed.

**Data availability statement** All data relevant to the study are included in the article or uploaded as online supplemental information. All relevant data to the study are included in the article or uploaded as supplementary information. Data are available upon reasonable request.

**ORCID iD**
Birgit Maria Vahlberg http://orcid.org/0000-0002-1508-1435

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
