## [Reviewer comments · BMJ Open]

ARTICLE DETAILS

TITLE (PROVISIONAL)	Potential effects on cardio-metabolic risk factors and body composition by short message service (SMS) guided training after recent minor stroke or transient ischemic attack: Post-hoc analyses of the STROKEWALK randomized controlled trial.
AUTHORS	Vahlberg, Birgit; Lundström, Erik; Eriksson, Staffan; Holmback, Ulf; Cederholm, Tommy

VERSION 1 – REVIEW

REVIEWER	Devasahayam, Augustine University Health Network
REVIEW RETURNED	27-Jul-2021

GENERAL COMMENTS	The authors have conducted a secondary analysis of data collected from a randomized controlled trial of patients who were treated for stroke at the Uppsala University Hospital, Sweden. The authors did not notice any benefits to sending short messages through mobile phones on body composition, cardiovascular risk factors, and self-perceived health for patients with stroke after discharge from the hospital. The findings from this study may have implications for clinical practice. But I think that it is important to provide a clear rationale for the connection between the healthcare system change (SMS intervention) and the biological outcomes in patients after stroke (body composition variables, blood lipids, HbA1c, and blood glucose) in the introduction. I read that relevant statistical analyses were completed to support the conclusions. However, there is no mention of whether the variables/data were checked for any missing data and outliers. I suggest adequate reporting of limitations of this study; for example, mentioning the role of confounding factors (which you were not able to control for) that interact with the study outcomes might help. Please explain why the 'mortality at one year' variable was collected from the medical records, and how it was analyzed. Please revise and report the 'patient and public involvement' section in the results. There are no meaningful details pertinent to the study aims in this section as of now. It's striking to see how the computerized 1:1 ratio randomization, stratified by gender, resulted in 37% females in your study cohort. I think it is important to further examine sex (biological) and gender
---

	(socio-cultural) differences in the outcomes, provided you have enough power to do sub-group analysis.
--	--

REVIEWER	Kontou, Eirini University of Nottingham, Division of Rehabilitation & Ageing
REVIEW RETURNED	11-Aug-2021

GENERAL COMMENTS	Thank you for your request to review this paper. This is well written and presents some secondary analyses from a previously reported RCT investigating the effects of text messaging supported training on physical activity/overall mobility - an interesting topic. However, I have some reservations and points for clarification. The article summary (p5) acknowledges the limitations of results as this is an explorative post-hoc study from a previously published RCT. It would be useful to state some reason why these analyses were not originally considered for inclusion in the other paper (reference number 12). On the 'strengths and limitations' section (p5), it seems that the limitations outweigh the strengths so I encourage the authors to consider identifying some additional strengths in relation to their present findings. Could you clarify what this paper adds in relation to the effects of the intervention? The Introduction section can benefit from further literature highlighting why cardio-metabolic risk factors and body composition are important for secondary prevention post TIA and minor stroke. I think the link between physical activity/exercising and the effects on the outcome assessments used for post-hoc analyses can be strengthened. Methods can be more succinct and avoid repetition of information presented in the main RCT paper (eg, randomisation, details of all baseline examinations - also reference on line 32, p8 does not seem correct). Results and analyses seem generally appropriate. Table 2 results can be further described in the text and the significant findings (p values) highlighted in bold so that they stand out for the reader. Please also clarify the use of correlational analyses - why needed and what this information adds to the aims of this paper. Overall the authors acknowledge the limitations of their findings as a post-hoc study, however there is no clear justification on why these analyses are not presented in the published RCT paper (Vahlberg et al, 2021; ref 12). This paper would be revised to better present its aims/findings and to be more explicit about what it adds in the existing literature.
--

VERSION 1 – AUTHOR RESPONSE

Reviewer 1

(Dr Augustine Devasahayam, University Health Network).

Remark 1: The findings from this study may have implications for clinical practice. But I think that it is important to provide a clear rationale for the connection between the healthcare system change (SMS intervention) and the biological outcomes in patients after stroke (body composition variables, blood lipids, HbA1c, and blood glucose) in the introduction.

Response: We have provided a more clear rationale for the connection between the effect of the intervention and the biological outcomes (see page: 4-5, line: 91-100)

Remark 2: I read that relevant statistical analyses were completed to support the conclusions. However, there is no mention of whether the variables/data were checked for any missing data and outliers.

Response: Missing values are now given in table 2-3. We have one potential outlier: The fat-free mass increased from 53.5 kg to 69.5 kg from the first to the second examination. This individual was omitted from analyses of body composition.

Remark 3: I suggest adequate reporting of limitations of this study; for example, mentioning the role of confounding factors (which you were not able to control for) that interact with the study outcomes might help.

Response: In the discussion (page 13, line: 287-288) we describe some confounding factors which we couldn't control for and that might have interacted with the study outcomes.

Remark 4: Please explain why the 'mortality at one year' variable was collected from the medical records, and how it was analyzed.

Response: Thanks for the remark. In page 9 (line: 183-184) we describe: "Mortality at one year was checked by reviewing the individual's medical records that is connected to the official death records".

Remark 5: Please revise and report the 'patient and public involvement' section in the results. There are no meaningful details pertinent to the study aims in this section as of now.

Response: Done (Page: 12-13, line: 267-271).

Remark 6: It's striking to see how the computerized 1:1 ratio randomization, stratified by gender, resulted in 37% females in your study cohort. I think it is important to further examine sex (biological) and gender (socio-cultural) differences in the outcomes, provided you have enough power to do sub-group analysis.

Response: We have expanded the statistical analysis with a sex specific between group analysis, which did not change the results (Page: 12, line: 252-254).

Reviewer: 2

(Dr. Eirini Kontou, University of Nottingham)

Remark 1: The article summary (p5) acknowledges the limitations of results as this is an explorative post-hoc study from a previously published RCT. It would be useful to state some reason why these analyses were not originally considered for inclusion in the other paper (reference number 12).

Response: Thanks for the remark. In the introduction (page: 4-5 and line: 91-100), we describe the reason why we performed these analyses and why they were not originally considered for inclusion in the other paper (ref. 12).

Remark 2: On the 'strengths and limitations' section (p5), it seems that the limitations outweigh the strengths so I encourage the authors to consider identifying some additional strengths in relation to their present findings. Could you clarify what this paper adds in relation to the effects of the intervention?

Response: Done. For example, in the 'strengths and limitations' section we describe that this paper is able to present results that highlights the importance of physical activity and how sedentary participants compared to those with moderate/high physical activity had an overall worse metabolic

risk-factor profile; e.g. higher body mass index, HbA1c, fat-mass and a lower HDL cholesterol at the time for stroke or TIA.

Remark 3: The Introduction section can benefit from further literature highlighting why cardio-metabolic risk factors and body composition are important for secondary prevention post TIA and minor stroke. I think the link between physical activity/exercising and the effects on the outcome assessments used for post-hoc analyses can be strengthened.

Response: We have extended the introduction with a more clear rationale why we did this study and why we report some variables that was not included in the originally study. Se remark 1.

Remark 4: Methods can be more succinct and avoid repetition of information presented in the main RCT paper (eg, randomisation, details of all baseline examinations - also reference on line 32, p8 does not seem correct).

Response: Thanks for the remark. We have deleted the description of the randomization and some of the details about the baseline examinations, to make the methods more succinct (see page: 6, line: 134-135). Unfortunately, we could not recognize which reference that doesn't seem correct.

Remark 5: Results and analyses seem generally appropriate. Table 2 results can be further described in the text and the significant findings (p values) highlighted in bold so that they stand out for the reader. Please also clarify the use of correlational analyses - why needed and what this information adds to the aims of this paper.

Response: We removed the correlation analyses and highlighted in bold the significant findings in table 2. We further describe the results presented in table 2 in the results section (page: 11, line: 229-233).

Remark 6: Overall the authors acknowledge the limitations of their findings as a post-hoc study, however there is no clear justification on why these analyses are not presented in the published RCT paper (Vahlberg et al, 2021; ref 12). This paper would be revised to better present its aims/findings and to be more explicit about what it adds in the existing literature.

Response: Beyond what is said in remark 1, already when we planned the study we decided to separate the results on physical function from those on body composition and cardio-metabolic risk factors. Retrospectively, we agree that due to the non-significant results of the latter objectives it could have been considered to integrate these results also in the major outcome paper.

VERSION 2 – REVIEW

REVIEWER	Devasahayam, Augustine University Health Network
REVIEW RETURNED	23-Sep-2021
GENERAL COMMENTS	The authors have addressed all of my concerns. Thank you.